# Microstructural, Mechanical and Corrosion Characteristics of Degradable PM Biomaterials Made from Copper-Coated Iron Powders

**DOI:** 10.3390/ma15051913

**Published:** 2022-03-04

**Authors:** Miriam Kupková, Martin Kupka, Andrea Morovská Turoňová, Renáta Oriňaková

**Affiliations:** 1Institute of Materials Research of SAS, Watsonova 47, 040 01 Košice, Slovakia; 2Institute of Experimental Physics of SAS, Watsonova 47, 040 01 Košice, Slovakia; kupkam@saske.sk; 3Department of Physical Chemistry, Faculty of Sciences, Pavol Jozef Šafárik University in Košice, Moyzesova 11, 041 54 Košice, Slovakia; andrea.morovska.turonova@upjs.sk (A.M.T.); renata.orinakova@upjs.sk (R.O.)

**Keywords:** Fe-Cu, coating, mechanical properties, corrosion characteristics, biodegradable, antibacterial function

## Abstract

Copper-containing iron-based materials have recently been recognized as potential biomaterials possessing antimicrobial ability. Since then, iron-copper systems have been prepared by different methods and investigated. This article is focused on PM materials made from composite powders. The powders, each particle of which consisted of an iron core and a copper shell, were prepared by electroplating. Test-pieces with copper contents of 0, 3.2, and 8 wt.% were fabricated by pressing and sintering from iron and composite powders. Some microstructural, mechanical, and corrosion characteristics of test-pieces were examined. Microstructures were composed of pores and iron grains with alloyed peripheral regions and copper-free cores. As the copper content in test-pieces was increased, their density and Young’s modulus decreased, and macrohardness, corrosion potential and corrosion current density increased. Likely causes of density and Young’s modulus reduction were higher porosity, low enough copper content, and compliant inclusions in stiff matrix. The increase in macrohardness was attributed to the precipitation hardening which prevailed over softening induced by pores. The increase in corrosion potential and corrosion current density was most likely due to the presence of more noble phase providing surfaces for a faster cathodic reaction.

## 1. Introduction

The search for ideal metals for medical implants is practically a never-ending quest.

Up until recently, almost all the attention of the biomaterials community was focused on developing bioinert, highly corrosion-resistant metals which should withstand aggressive environments and should persist in the human body forever. However, since the concept of in-patient corroding implants emerged, more and more attention has been paid to developing biodegradable, quite corrodible metals, which should dissolve in body fluids and should leave the human body via physiological pathways.

In theory, biodegradable implants should stay in the human body only until the health problem is adequately fixed and should gradually degrade away thereafter. To do their job well, biodegradable implants must be made from materials which have to satisfy complex mechanical, chemical, and medical requirements. For example, there is an imperative need for degradation rate controllability and adjustability.

Iron, iron alloys, iron-matrix composites, and iron compounds are some of the materials under consideration for the potential use in biodegradable medical devices [1,2].

Not so long ago, the concept of self-sanitizing implants emerged. Additionally, since then, the research in the field of biodegradable metallic materials has begun to focus a little more on biodegradable metallic materials with antimicrobial properties (see review papers [3,4] and references therein).

People have fabricated various binary iron-based systems containing silver [5,6,7], copper [7,8,9], or zinc [10,11], the most common alloying elements with good antibacterial properties, though often without any intention to produce self-disinfecting products.

The copper-bearing iron-based binary materials were fabricated in a variety of ways. For example, nanocomposite particles, which were prepared by mechanical alloying [7] or by explosion [12], were subsequently cold sintered (consolidated by application of very high pressure), or conventional mixtures of elemental iron and copper powders were consolidated by selective laser melting [8] and by microwave sintering [9].

This article is focused on iron-copper materials prepared by conventional pressing and sintering from composite powders. Each particle of these powders consisted of an iron core and a copper shell. To gain preliminary insight into degradation (corrosion) behavior of such materials in a (pseudo)physiological environment, a medium mimicking the composition and pH value of body fluids for corrosion tests was chosen, namely, Hanks’ solution, since it was designed to create test environments as close as possible to the environment in the human body [13].

## 2. Materials and Methods

Water atomized iron powder ASC 100.29 from Höganäs AB (Höganäs, Sweden), was used as a starting material.

Iron particles with diameters larger than 63 μm and smaller than 180 μm, sieved from the raw powder, were used as substrates for the galvanostatic deposition of copper from an aqueous electrolyte containing 0.25 M sodium citrate and 0.06 or 0.1 M copper sulphate.

The electroplating was performed in an electrolytic cell with a fluidized bed electrode. The electrolytic cell was comprised of inner and outer cylindrical vessels. The interior of the inner vessel defined the cathode compartment of the cell, and the space between the inner and outer vessels defined the anode compartment of the cell. The inner vessel had an orifice in its side wall. This orifice was sealed with a porous diaphragm providing suitably sized pores that allowed the electrolyte to pass from one vessel to the other, but prevented powder particles from passing though the diaphragm.

The inner vessel was loaded with 30 or 20 g of powder and there was a plating electrolyte in both vessels of the cell. The fluidized bed of powder particles in electrolyte was generated and maintained in the inner vessel by means of an intensive circular stirring at a rate of 1100 rpm. The current feeder for the fluidized bed cathode was a stainless-steel plate placed in the inner vessel, the copper plate placed in the space between the inner and outer vessels served as an anode. During electroplating, a current of 1 A was applied which flowed for 30 min.

The schematic sketch and parameters of the electrolytic cell were published in [14].

After the electroplating was completed, coated particles were rinsed in distilled water and dried. The content of copper deposited on iron particles was determined after dissolution of coated particles by means of atomic absorption spectrometry (AAS).

From the coated powders, suitably shaped compacts (prismatic bars) were produced by uniaxial die pressing under the pressure of 600 MPa. No lubricant was used. The green compacts were then sintered in a laboratory carbolyte furnace. During the sintering process, the furnace was heated up at a rate of 10 °C/min, held at the sintering temperature of 1120 °C for 60 min, and cooled down at a rate of 10 °C/min to room temperature. The reducing sintering atmosphere consisted of H_2_ and flowed at a rate of 4 L/min.

To prepare the samples for metallographic examinations and hardness or microhardness measurements, selected sintered specimens were cut, mounted, ground, polished and, if needed, also nital etched. The microstructure of samples was studied using a light microscope Olympus GX71 (OLYMPUS Europa Holding GmbH, Hamburg, Germany) (LM), and scanning electron microscope coupled with the energy dispersive spectrometer, JEOL JSM-7000F (Jeol Ltd., Tokyo, Japan), with EDX INCA (SEM).

Information about macrohardness, microhardness and stiffness (Young’s modulus) of investigated specimens was drawn from the data provided by indentation tests and flexural vibration tests.

The macro-indentation tests were performed on the Wolpert Wilson 432 SVD hardness testing machine (ITW Test and Measurement GmbH, Esslingen, Germany) with a Vickers diamond indenter. The Vickers indenter was pressed into the surface of a test-piece at a load of 10 kgf. The size of the impression left on the specimen surface by the indenter was measured with the aid of a microscope and the Vickers hardness number HV10 was calculated. The indentations were repeated ten times for each sample. The resultant value is an arithmetic average from the repeated measurements and represents the effective macroscopic hardness of the material as a whole (skeleton plus pores).

The Vickers microhardness was measured using a LECO LM-700AT microhardness tester (LECO Instrumente Plzeň, s.r.o., Plzeň, Czech Republic) equipped with a micro-indenter with Vickers tip, by applying a load of 10 gf. For each sample, 300 indents were performed at different locations throughout the solid regions between pores to study the distribution of hardness values throughout the skeletal material and to determine the range of these values, which reflects the heterogeneity of microstructure of the examined material.

After impact excitation of vibration of prismatic bar, the natural frequency of fundamental flexural mode was detected using the Buzzosonic 5.96 equipment (BuzzMac International, LLC, Portland, ME, USA). Using the knowledge of mass, geometry, and fundamental bending frequency of a test-piece made it possible to calculate the (effective) Young’s modulus, that is in essence the stiffness of test-piece’s material.

The electrochemical corrosion parameters, namely, corrosion potentials and corrosion current densities, were extracted from the data provided by the potentiodynamic polarization tests.

Polarization measurements were performed at 37 °C in a standard three-electrode, one-compartment electrochemical cell containing Hanks’ solution as an electrolyte. The three electrodes were connected to the Autolab PGSTAT 302N potentiostat (Metrohm AG, Herisan, Switzerland) which controlled the voltage between the working electrode and the reference electrode while measured the resulting current between the working electrode and counter electrode.

A test-piece was coated with a chemically inert, electrically insulating coating, except for a small surface area of 1 cm^2^ which remained uncoated, accessible for the electrolyte and served as a working electrode. The Ag/AgCl (3 M KCl) electrode was used as a reference electrode, and the counter electrode was a platinum foil.

After immersion in the electrolyte, the working electrode was allowed to corrode without any external intervention until the voltage between the freely corroding working electrode and the reference electrode reached the steady-state value—the open-circuit potential or corrosion potential *E_corr_* of the working electrode material in Hanks’ solution.

After nearly 1 h of stabilization, the voltage between working and reference electrodes was varied from −800 to −200 mV at a scanning rate of 0.1 mV/s and the resulting current between working and counter electrodes was recorded. The obtained data were converted into the current density at the working electrode, *i*, and potential, *E*, of the working electrode versus reference electrode. Plots of log|*i*| versus *E* (i.e., Tafel plots) were constructed and the Tafel extrapolation method was used to extract the value of corrosion current density. *i_corr_*, from these plots.

## 3. Results and Discussion

During the electroplating, the original iron powders transformed into composite ones consisting of powder particles which were at a glance almost entirely covered with a copper.

Figure 1 exhibits typical bare and coated powder particles. When comparing Figure 1a–d, it can be seen that the surface structure changed slightly at micron and submicron levels after deposition of copper. While the surface of bare iron particles is relatively smooth, the surface of copper-coated particles became grainier and rougher.

Each individual particle consisted of an iron core and almost compact copper shell. Figure 2 shows cross-sections of typical copper-coated iron particles. A relatively uniform coating layer of copper on the surfaces of iron particles is quite visible.

A quantitative analysis revealed the copper content of 3.2 wt.% in the group of powders processed in larger batches in a less concentrated electrolyte and of 8 wt.% in the group of powders processed in smaller batches in a more concentrated electrolyte.

A typical microstructure of green compacts fabricated from copper-coated iron powders by uniaxial die pressing is presented in Figure 3. Copper layers are visible along the boundaries between deformed iron particles. However, the axial sliding during die compaction led to the wear of surfaces of adjacent particles and removed part of the copper from the contacts between particles to inter-particle voids nearby, which can be seen in Figure 3.

A typical microstructure of sintered materials is presented in Figure 4 and Figure 5.

In Figure 4, the photos of non-etched sintered specimens are presented to demonstrate the evolution of porosity with increasing copper content, from a pure iron (Figure 4a), through to a specimen with 3.2 wt.% Cu (Figure 4b), to a specimen with 8 wt.% Cu (Figure 4c). As the content of copper was increased, the size of pores also visibly increased.

In Figure 5, chemically treated cross sections of sintered specimens with 3.2 wt.% Cu (Figure 5a) and 8 wt.% Cu (Figure 5b) are presented. The specimens were etched with nital to make the distribution of Cu atoms throughout the specimen cross section visible. The regions containing copper were attacked with the etching agent and became darker, while the pure iron regions were not attacked and remained bright. For all specimens, small grains were completely dark, while larger grains had bright cores and dark peripheral regions. Thus, the microgradient structure was observed. Isolated regions occupied by a pure metallic copper were found along former powder particle boundaries in specimens with 8 wt.% Cu and the peripheral regions were quite dark in these materials, while no metallic copper phase was observed in specimens with 3.2 wt.% Cu and grain peripheral regions were not so dark there.

Figure 6 shows a typical concentration profile of copper recorded using an EDX line scan across a sintered material region originated in a single copper-coated iron particle and contacts with its neighbors. It can be seen that a gradient was created during sintering from high concentration at copper coating locations to lower concentration at the grain interiors.

Figure 7 shows adjoining copper-bearing and copper-free grain regions. As can be seen in this image, which was taken at higher magnification, there are many submicron areas occupied by copper (bright spots) on the surface of copper-bearing material and practically none on the surface of copper-free material.

From the iron-copper equilibrium phase diagram [15] it can be seen that the limit of solubility of copper in iron is about 7.2 wt.% at the sintering temperature of 1120 °C. Thus, it was expected that in specimens with 3.2 wt.% Cu, such small amount of copper, subceeding the solubility limit, should completely dissolve into iron particles, while in specimens with about 8 wt.% Cu such large amount of copper, exceeding the solubility limit, should not dissolve into iron particles completely, some amount of copper should remain along the iron particle boundaries. This might explain the presence of metallic copper in specimens with 8 wt.% Cu (Figure 5b) and its absence in specimens with 3.2 wt.% Cu (Figure 5a).

Since the sintering was performed at 1120 °C, which is above the melting point of copper (1083 °C), the copper-based melt was present during sintering, either temporarily or permanently.

Due to the large surface tension of solid iron, a moderate surface tension of molten copper, and a low interfacial tension between solid iron and molten copper [16], the molten copper efficiently wets the surface of solid iron. Hence, when the temperature of sintered specimen exceeded the melting point of copper, copper melt and began rapidly flowing along surfaces of iron particles and into boundaries of iron grains within these polycrystalline particles [17]. This rapid spreading of molten copper over iron grains was accompanied and then followed by a much slower diffusion of copper atoms from periphery to the interior of iron grains [18]. All these processes caused the expansion of iron particles and left pores at the sites vacated by the copper, which affected the porosity characteristic of the specimen (Figure 4).

To demonstrate the trend in porosity evolution, the value of porosity was calculated for a three-phase material consisting of iron, copper, and voids using the measured density of real sintered material with the given copper content, densities of pure iron and pure copper, and their weight fractions in the given specimen (Table 1). The calculated porosity, which approximates the porosity of real specimens, increased as the copper content was increased.

In a specimen held for 60 min at 1120 °C, theoretically, the front of molten copper should travel from hundreds to thousands of microns along iron grain boundaries [17] while the front of solid-solution formation should move a few microns from the surface of the iron grain towards its center [18]. Thus, during an hour at the peak temperature, the copper should dissolve within the several microns thick peripheral regions of iron grains, forming a substitutional solid solution of copper in iron. Thus, a specimen with a gradient microstructure of larger iron grains should be created with copper-free cores surrounded by peripheries formed from a solid solution of copper in iron at the sintering temperature, or from a dual-phase iron-copper alloy at room temperature. Smaller grains should be formed entirely from a solid solution or from a dual-phase alloy (Figure 5 and Figure 6).

As the specimens were cooled down from 1120 °C to room temperature, the solubility of copper in iron dropped sharply to practically zero at room temperature [15]. Thus, the solid solution in peripheral regions of grains became supersaturated, that is thermodynamically unstable. As the copper and iron are not soluble in each other at room temperature, the copper tended to precipitate from the supersaturated solid solution.

Due to the production process based on the atomization of molten metal by jets of high-pressure water, individual particles of water-atomized iron powder contain many microscopically small pores, though the total porosity of particles is very low [19,20,21]. During the cooling of sintered specimens, these pores could be clogged by copper precipitating from a supersaturated solid solution. The submicron copper spots visible in Figure 7 are most likely the copper precipitates in these pores.

Besides the precipitates filling the “relict” intragrain voids, it is expected that there are also conventional bulk copper precipitates nucleated and growing within the iron lattice, since the copper does so to a certain extent naturally when the specimen is passing the cooling zone of the sintering furnace at usual rates [22,23,24].

A commonly accepted view is that bulk precipitation started with a homogeneous nucleation of copper clusters and the crystal lattice of precipitates changed from body-centered-cubic (bcc) at small sizes to face-centered-cubic (fcc) at large sizes through an intermediate martensitic stage [25]. The bulk precipitates are commonly very tiny, thus the individual precipitates are beyond the resolving power of microscopes used.

PM materials produced by conventional pressing and sintering are naturally porous. The presence of pores has a remarkable impact on material’s properties. Relying on a significant number of models proposed [26], it is believed that the stiffness and hardness decrease from those of the fully dense material to practically zero as the porosity is increased from zero porosity to the porosity of freely settled powder. Additionally, for a given value of porosity, the property of porous material is proportional to the property of the skeletal material.

Mechanical properties of skeletal Fe-Cu materials are affected by the presence of precipitates. It was expected that an increase in copper content caused an increase in the number of copper precipitates.

As regards the material’s elastic response to applied mechanical loads, copper precipitates represent a more compliant phase built in a stiffer iron matrix. Thus, the increasing copper content reduces the stiffness of fully dense Fe-Cu materials, that is, it reduces their effective modulus of elasticity (Young’s modulus) [27]. Hence, the stiffness (Young’s modulus) of investigated specimens should decrease with increasing copper content both due to increasing specimen’s porosity and due to decreasing Young’s modulus of skeletal material. Experimental results agreed with these predictions (Table 1).

In issues involving the material’s inelastic response to applied mechanical loads, very small copper precipitates dispersed in an iron matrix block the movement of dislocations. This increases the material’s hardness [28].

With increasing copper content, the macroscopic hardness of investigated specimens should decrease due to increasing specimen porosity and it should increase due to increasing hardness of skeletal material. Measured macroscopic hardness increased with increasing copper content (Table 1). This indicates that the hardening of skeletal material induced by precipitates prevailed over the softening of the whole material induced by pores.

To quantify how hard the skeletal material is at different locations on its surface and to estimate the range of occurring microhardness values which reflects the skeletal material heterogeneity, the specimen cross section was indented with a micro-indenter at numerous different locations throughout the solid regions of the specimen cross-section between the pores. The distribution of microhardness for typical specimens is presented as cumulative relative frequency versus hardness number plots in Figure 8.

When the indents are distributed more or less evenly throughout the surface, the value of cumulative relative frequency for a given hardness number can be considered as the fraction of surface on which the microhardness is equal to or less than the given hardness number.

The observed hardness values ranged from about 50 to about 200 HV 0.01 for iron specimens, from about 50 to about 300 HV 0.01 for specimens containing 3.2 wt.% Cu and from about 50 to about 400 HV 0.01 for specimens containing 8 wt.% Cu. The range of measured microhardness values increased when the content of copper in the specimen was increased. The maximum measured value of hardness also increased with increasing copper content.

Since the range of local values of concentration of copper becomes wider and the maximum local copper concentration becomes higher with increasing total copper content, the above results indicate that the microhardness of skeletal material increases at locations where the local concentration of copper is increased.

Potentiodynamic polarization measurements and porosity estimations revealed that the corrosion potential, corrosion current density, and total porosity increased as the content of copper in specimens was increased (Figure 9, Table 2). Since the previous study [29] revealed that the corrosion potential of sintered pure iron decreased when its porosity was increased, it is assumed that it is the presence of copper with its higher electrochemical nobility and faster oxygen reduction on its surface which decisively affects the corrosion of sintered iron-copper materials.

To gain an insight into the causes of corrosion behavior of materials investigated, a simplified version of reality was considered which allowed some qualitative predictions to be made.

To reduce the real problem to the problem manageable by means of mathematical analysis or deductive reasoning, some simplifying assumptions have been made. The iron-copper composite was used as an approximation to real sintered material. Corrosion reactions were narrowed down to one anodic and one cathodic reaction. These two reactions were the oxidation of iron atoms to ferrous ions, and the reduction of molecular oxygen in presence of water to hydroxide ions. Both reactions were assumed to be under activation control. Ohmic potential drops were assumed to be negligible both in solids and in solution.

When the above-mentioned assumptions hold true and the potential difference between the composite iron-copper electrode and the reference electrode is set at a value of corrosion potential of pure iron, anodic and cathodic current densities balance out at each point on the iron surfaces, but there is no anodic, only cathodic current density at each point on copper surfaces of the iron-copper electrode. Thus, there is a net cathodic current between the working and counter electrodes. To achieve a steady state with no net current entering or leaving the composite electrode, which corresponds to the spontaneous corrosion processes ongoing on the surface of composite, it is necessary to speed up the anodic reaction and to slow down the cathodic reaction. This means that the potential of the iron-copper electrode must be displaced from the value of corrosion potential of iron towards more positive values. The greater the surface area occupied by copper relative to that occupied by iron, the greater the potential displacement is needed.

Thus, the corrosion potential should increase with increasing content of copper in iron-copper materials. This prediction agrees with experimental data (Table 2).

The exchange current density for oxygen reduction reaction is more than ten times higher on copper surfaces than it is on iron surfaces [30,31]. The higher cathodic current density on copper surfaces may more than compensate for reduction in cathodic current density on iron surfaces caused by a slight increase in corrosion potential due to the copper present in minute amounts in the iron. Thus, the corrosion current per unit surface area can be lower on a pure iron surface than it is on an iron-copper composite surface with a low enough copper-to-iron area ratio.

The increase in current density with increasing copper content was observed experimentally (Table 2).

## 4. Conclusions

An interest in Fe-based Cu-containing materials has recently increased in connection with the emergence of the concept of self-disinfecting biodegradable implants, since these materials can biodegrade and can destroy pathogenic bacteria. Several types of binary and multicomponent systems have been investigated. Apart from the medical issues, various aspects of physics, mechanics, chemistry, and electrochemistry of these materials have been studied.

Unlike previous works dealing with iron-copper materials prepared by unconventional method (cold sintering) from unconventional powders (nanocomposite powder particles), or prepared by unconventional methods (microwave sintering, selective laser melting) from conventional powders (mixtures of elemental iron and copper powders), this article has focused on materials prepared by a conventional method (pressing in a die and sintering in a classical furnace) from a slightly unconventional powder (composite powders, each particle of which consisted of an iron core and copper shell).

The use of composite powders provided more homogeneous distribution of copper in the green compacts, and sintering at high temperature resulted in materials consisting of iron grains having alloyed peripheral regions and almost pure iron cores. The alloyed peripheral regions have very nearly the same thickness in all grains.

As expected, the stiffness of obtained materials (quantified by their Young’s modulus) decreased when the copper content was increased. The variation of macroscopic hardness could not be estimated, but the experiment has shown that the hardening of skeletal material induced by copper precipitates prevailed over the softening of whole material induced by pores so that the macroscopic hardness increased as the copper content was increased.

The corrosion potential increased with increasing content of copper, which was expected due to the presence of a more noble metal. It could not be predicted whether the (apparent) corrosion current density would increase or decrease. The experiment has shown that the (apparent) corrosion current density increased when the copper content was increased. Thus, under given experimental conditions, the concentration of copper was low enough so that the corrosion potential was displaced relatively slightly, which slowed down the cathodic reaction on iron surfaces only negligibly, and the cathodic reaction, which was faster on copper surfaces than it was on iron surfaces, was fast enough on copper surfaces to prevail its slowing down on iron surfaces.

Copper-coated iron powders, with a possibility to choose the thickness of copper shells, seem to be a promising raw material for fabricating macroscopically homogeneous, eventually biodegradable, iron-copper sintered parts with tuned mechanical and electrochemical properties. However, it is necessary to gain a more comprehensive knowledge to be capable of realizing this possibility.

## Figures and Tables

**Figure 1 materials-15-01913-f001:**
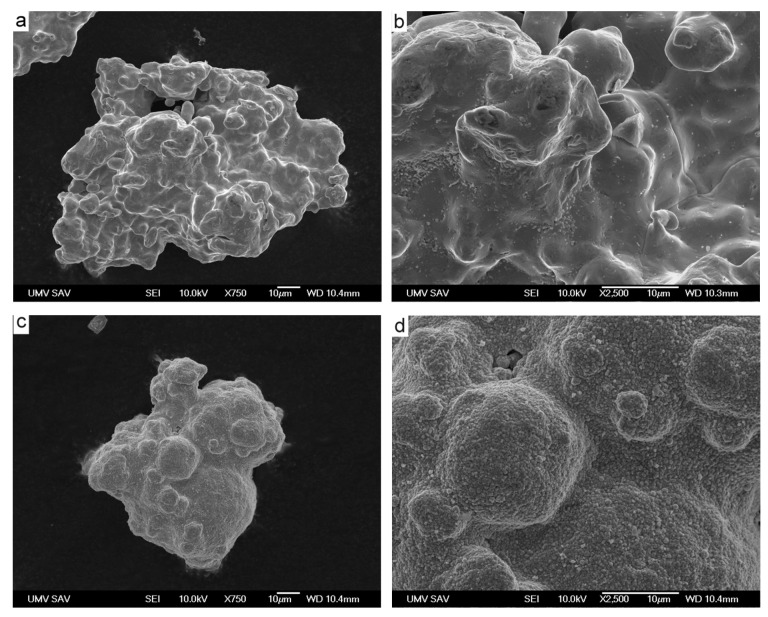
Typical bare (**a**,**b**) and copper-coated (**c**,**d**) particles of water-atomized iron powder (ASC 100.29, Höganäs AB). The content of copper in presented powders is 0 wt.% (**a**,**b**) and 8 wt.% (**c**,**d**), SEM.

**Figure 2 materials-15-01913-f002:**
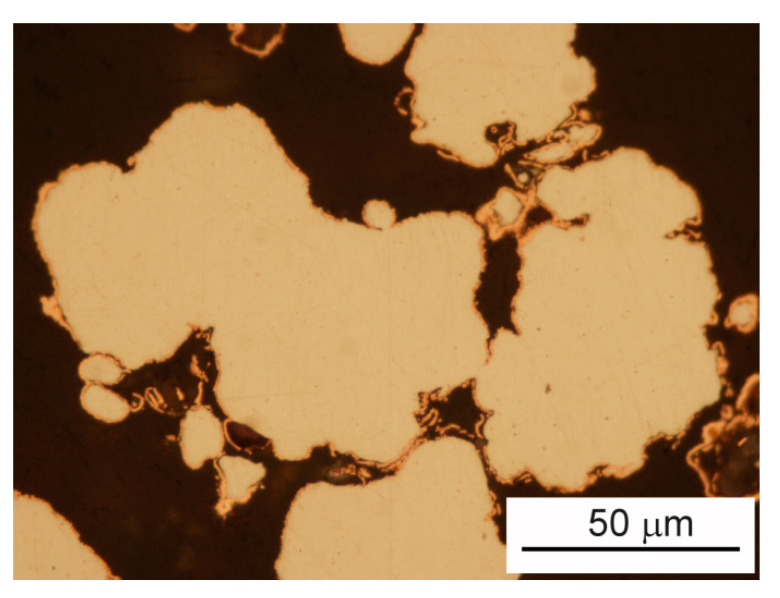
Cross-sections of typical copper-coated iron particles. The content of copper in presented powder is 8 wt.%, LM.

**Figure 3 materials-15-01913-f003:**
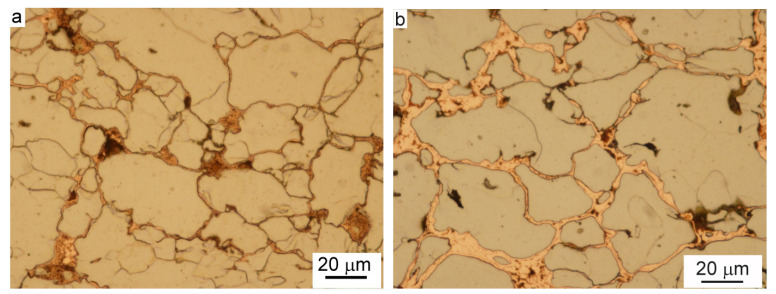
A typical microstructure of compacted copper-coated iron powders. The specimens presented were fabricated by applying a pressure of 600 MPa on copper-coated iron powders containing 3.2 wt.% Cu(**a**) and 8.0 wt.% Cu (**b**), nital etched, LM.

**Figure 4 materials-15-01913-f004:**
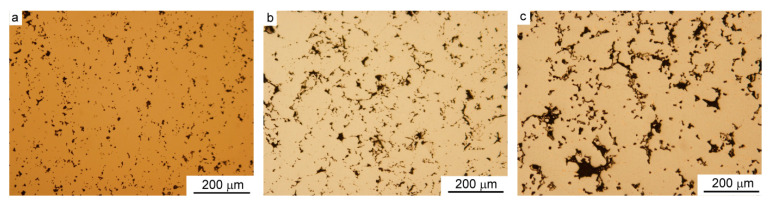
Pores in sintered materials: (**a**) Fe, (**b**) Fe-3.2 wt.% Cu, and (**c**) Fe-8.0 wt.% Cu, LM.

**Figure 5 materials-15-01913-f005:**
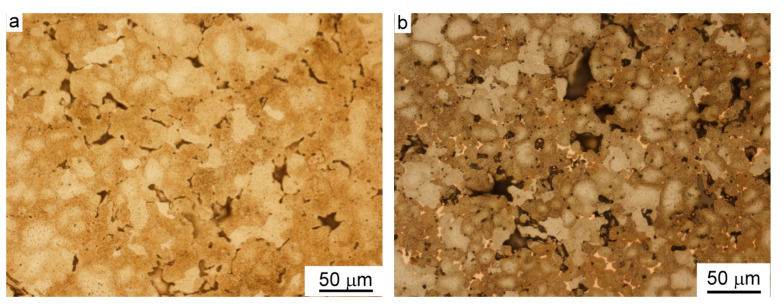
Microstructure of sintered materials: (**a**) Fe-3.2 wt.% Cu and (**b**) Fe-8 wt.% Cu, nital etched, LM.

**Figure 6 materials-15-01913-f006:**
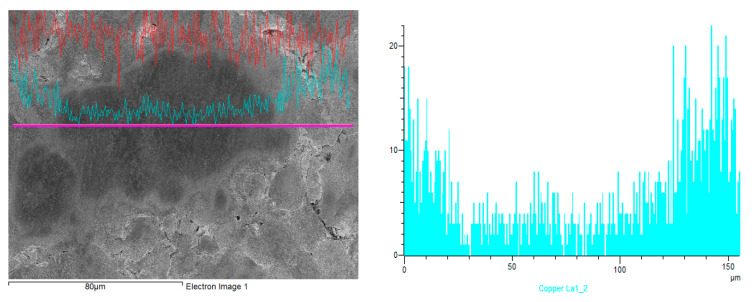
A region of sintered specimen originated in a single coated particle and contacts with its neighbors (**left**), and a copper concentration profile obtained from EDX line scan (**right**). The scan was taken from left to right side along the line marked in the left figure. The specimen with 3.2 wt.% Cu is presented, SEM.

**Figure 7 materials-15-01913-f007:**
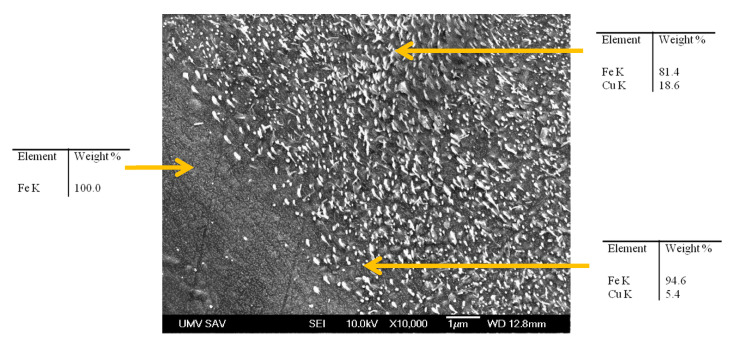
Adjoining copper-bearing and copper-free grain regions, and the elemental composition identified by EDX at different locations on the grain section. The specimen with 3.2 wt.% Cu is presented, SEM.

**Figure 8 materials-15-01913-f008:**
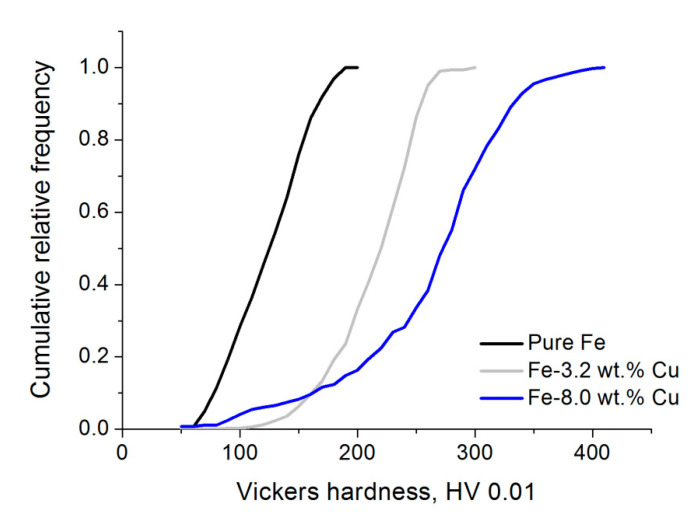
Cumulative relative frequency of occurrence as a function of hardness number.

**Figure 9 materials-15-01913-f009:**
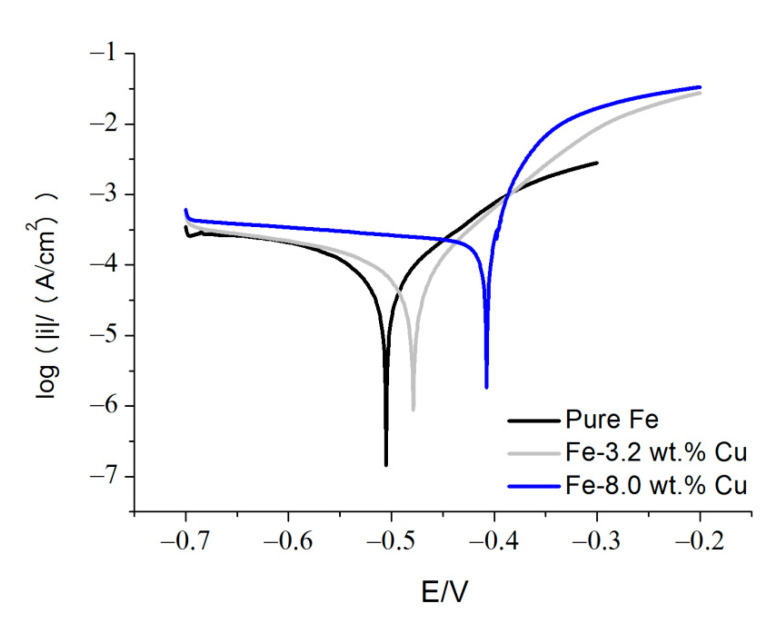
Tafel plots for typical sintered iron and iron-copper samples.

**Table 1 materials-15-01913-t001:** Physical and mechanical parameters of sintered iron and iron-copper materials.

	Pure Fe	Fe-3.2 wt.% Cu	Fe-8.0 wt.% Cu
Density (g/cm^3^)	7.19 ± 0.1	6.98 ± 0.1	6.82 ± 0.2
Approximate porosity (%)	8.6 ± 0.1	11.5 ± 0.2	14.1 ± 0.6
Vickers hardness (HV 10)	63 ± 3	98 ± 3	161 ± 5
Young’s modulus (GPa)	154.2 ± 0.3	124.7 ± 0.4	110.1 ± 0.6

**Table 2 materials-15-01913-t002:** Corrosion parameters for typical sintered iron and iron-copper samples.

	Pure Fe	Fe-3.2 wt.% Cu	Fe-8.0 wt.% Cu
*i_corr_* (µA/cm^2^) *E_corr_* (mV)	48	57	100
	−505	−479	−407

## Data Availability

The data presented in this study are available on request from the corresponding author.

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
