# Peer review of "Microstructural, Mechanical and Corrosion Characteristics of Degradable PM Biomaterials Made from Copper-Coated Iron Powders"

_materials, 2022, doi:10.3390/ma15051913_

Round 1
Reviewer 1 Report
The article raises the preparation of biomaterials composed by powder metallurgy. This work is interesting, but it needs to be improved before the publication on this journal.
For the density of composite, the authors followed any international guideline, or an internal procedure please explain.
The authors show that some mechanical properties (Tensile strength and Vickers hardness) of the composites are improved by increasing the number of pores. It is not very clear why this occurs, the authors could improve this explanation
The authors should include a structural analysis by x-ray diffraction and to include study the size of crystals.
The authors reported preparation of composite biomaterials. Authors should include some characterization that demonstrates that this property
Author Response
After careful consideration of the reviewers' comments, we have responded as follows:
We did not change the Title and the character of Introduction, since the degradation (corrosion) of investigated materials was tested in Hanks' solution which mimics the environment in the human body. The biodegradability of prospective materials is usually examined in vitro by degradation (corrosion) in simulated body fluids before the more expensive in vivo experiments will be performed.
We removed parts dealing with the tensile and three-point-bend test (description of method, result, discussion), since we cannot obtain curves recorded during experiment within a reasonable time.
We added LM and SEM images of bare and coated particles, of cross sections of coated particles, of green compacts, of microstructures of sintered specimens with results of EDX line scan and elemental analysis, with corresponding discussion.
We added information on corrosion of pure sintered iron with different values of porosity.
We added references to Introduction and to text where it was needed.
We added parts dealing with measurements of microhardness (method, results, discussion) to demonstrate a possible connection between the microhardness and local copper concentration.
Reviewer 2 Report
Dear Authors,
below I wrote my comments on your paper. Please, try to improve your manuscript to show obtained results but under another title – more fitted to the content. By now, your explanation in the paper is readable, but there is a lot of debatable content without support in the results.
Introduction, as well as the paper title, suggest that the paper contains the data for biodegradable biomaterial. In fact, there are only some basic metallographic preparation, basic mechanical tests performed of tensile strength, flexural strength, hardness and Young modulus and corrosion tests with the use of the potentiodynamic method. The chosen set of investigation methods is correct but not in the paper with the proposed title and – as a consequence – introduction. In the present form, the Introduction section needs rewriting because does not introduce the main text of interesting materials of Cu coated Fe powders – mechanical and corrosion resistance. The reader does not know what are the positive aspects of coating the granules of powder in comparison to similar materials manufactured with the use of the conventional PM method.
Research design needs supplementary material. First of all the technology description. It is not clear if the 3.2wt.% is a product of embedding 30g of Iron powder to the vessel or 20g. Why such a small change of copper content when adding/removal of 1/3 powder mass? Is the technology repeatable?
Powder characterization:
In my opinion, the way more attention should be paid to the proper characterization of powders before and after the technological process of coating. Just the use of big (63 - 180µm) grains of Iron as the biomaterial is in my opinion not correct. For common mechanical tests it would be ok, but for degradable biomaterial is something wrong with this proposition. The coated powders were not investigated at all. Does the iron powder oxidize when rinsed with distilled water? (line 80) In my opinion, it should corrode with just short contact with water. Does the copper coat the full grains of Iron powder? What was the thickness of the copper coating? Was the coating uniform? Why there is no evidence of chemical or phase composition of Iron powder as well as coated ones? Does the electrochemical reaction products are embedded/caught in the copper coating? Was the copper coating porous? Was the copper coating oxidized?
Sintering:
Please show/describe the sintered specimens' dimensions. The 600MPa of pressure means the pressure of the press or it is pressing pressure for the samples? It is quite high for Iron or Copper anyway. The sintering temperature of 1120C was – as written – above the melting point for Copper. So, it was sintering with a liquid phase. Such a fact made the experiment even more complex to appraise. The Authors gave a very detailed description of possible phenomena but the ones taking place in those particular samples were not investigated. All the results of the microstructure are shown only as metallographic pictures. In my opinion, it is a way too small possibility to appraise the phenomena taking place during sintering. Why there is no SEM/EDS or other methods for confirming the presence of copper, copper–iron solution, or copper precipitates (f. ex. TEM or XRD) as the Authors wrote. Again: Do the samples oxidize during sintering?
Results:
The Authors gave possible answers of observed/measured data for their samples but without deeper investigations. In terms of a scientific paper, I appraise this as a main drawback of the paper. It is hard to believe that material based on iron, with the addition of copper up to 8 wt.%, with quite high porosity (up to 14%!) has 400 MPa of Tensile Strength and 780 MPa of Flexural Strength. The Authors have not shown the curves of those tests so we cannot appraise them. In my opinion, the stressing curves could be interesting as the investigated materials were not common.
How does the porosity influence the mechanical parameters of the samples? Precipitation hardening has to be conducted within very strict parameters of time and temperature. Why do the Authors suggest – without investigations – that a similar process took place in their samples and this is the main reason for obtaining such high mechanical parameters? Again – the lack of basic characterization of powders and sintered samples does not allow for such discussions.
Corrosion resistance:
How do look the Tafel plots for Iron sinters with different porosity? How do the (proposed by the Authors) Copper precipitates influence the corrosion behavior?
Concluding remarks:
The title does not match the content because there were not investigated the degradation of biomaterial.
The introduction section should be improved by removing biomaterials references and adding comparable materials properties to be seen.
The technology section description needs to be improved.
Basic characterization of powders and sintered powders should be included as well as precipitation of copper should be shown (f. Ex. the precipitation particles dimension)
The mechanical test results should be shown on the diagrams.
Corrosion test should be shown as a comparison versus porosity of pure Iron samples with different porosity because probably it can be shown that the porosity of pure Iron has a bigger influence than Copper content and Copper state (coating, Capillary effect, or precipitates in Iron).
Author Response

(The authors gave the same response as above.)

Reviewer 3 Report
The work presented is commendable. However, it seriously lacks experimental justification.
1) SEM pics and EDX area mapping/line scanning should be provided for composite (copper coated iron) powder, their sinterred state (to show the diffusion induced copper distribution in the iron powder) to justify the chemical composition, core/shell elemental distribution.
2) SEM and EDX analysis of corrosion product and corrosion surfaces to justify the corrosion result.
Author Response

(The authors gave the same response as above.)

Round 2
Reviewer 3 Report
Authors wrote that they removed tensile and bending test results. However, tensile test description, stiffness of the materials is extensively mentioned in abstract, methodology and result and discussion.
Hardness test method mentioned twice with different machine and parameters.
Author Response
Response to Reviewer 3
Authors wrote that they removed tensile and bending test results. However, tensile test description, stiffness of the materials is extensively mentioned in abstract, methodology and result and discussion.
Response: All text dealing with tensile and bending tests was indeed removed. The stiffness of materials is in essence their Young's modulus, which was measured using the vibrations of bar-shaped specimens, not tensile tests. For the sake of clarity, we explicitly expressed the connection between stiffness and Young's modulus at various locations in article.
Hardness test method mentioned twice with different machine and parameters.
Response: One equipment was used for measuring the macroscopic hardness by applying a higher load, the other was used for measuring the microhardness by applying a very low load.